# Van der Pol-informed Neural Networks for Multi-step-ahead Forecasting of Extreme Climatic Events

**Anurag Dutta**
GCETT Serampore, India
anuragdutta.research@gmail.com

**Madhurima Panja**
IIIT Bangalore, India
madhurima.panja@iiitb.ac.in

**Uttam Kumar**
IIIT Bangalore, India
uttam@iiitb.ac.in

**Chittaranjan Hens**
IIIT Hyderabad, India
chittaranjan.hens@iiit.ac.in

**Tanujit Chakraborty**
Sorbonne University Abu Dhabi, UAE
tanujit.chakraborty@sorbonne.ae

## Abstract

Deep learning has produced excellent results in several applied domains including computer vision, natural language processing, speech recognition, etc. Physics-informed neural networks (PINN) are a new family of deep learning models that combine prior knowledge of physics in the form of high-level abstraction of natural phenomena with data-driven neural networks. PINN has emerged as a flourishing area of scientific computing to deal with the challenges of shortage of training data, enhancing physical plausibility, and specifically aiming to solve complex differential equations. However, building PINNs for modeling and forecasting the dynamics of extreme climatic events of geophysical systems remains an open scientific problem. This study proposes Van der Pol-informed Neural Networks (VPINN), a physics-informed differential learning approach, for modeling extreme nonlinear dynamical systems such as climatic events, exploiting the physical differentials as the physics-derived loss function. Our proposal is compared to state-of-the-art time series forecasting models, showing superior performance. The codes and dataset used for the experiments are made available at https://github.com/mad-stat/VPINN.

## 1 Introduction

Recently with the growing threat of climate change, there has been a global surge in the emergence and intensity of extreme climatic events [20]. The abrupt occurrence of devastating natural calamities like earthquakes, hurricanes, droughts, and floods poses significant challenges to human lives, infrastructure, ecosystems, and economies. Such tragic real-world extreme events include the 2023 Marrakesh earthquake in Morocco, the earthquake in Turkey, the cyclone Biparjoy in India, the recent flood in Spain, and the oceanic rouge wave near Newfoundland. To address the severity of these extreme climatic phenomena, accurate forecasts of these abrupt events have become imperative to government agencies for disaster preparedness and adaptation.

In recent decades, researchers have increasingly delved into the study of extreme events using dynamical systems and data-driven machine learning techniques, although typically in isolation

37th Conference on Neural Information Processing Systems (NeurIPS 2023).

[10, 9, 5, 4]. Accurate prediction of extreme events through physics-based dynamical systems relies heavily on understanding the system's inherent dynamics. In contrast, machine learning models, such as reservoir computing network (RCN) [16], long short-term memory (LSTM) [7], bidirectional LSTM (Bi-LSTM) [17], Prophet [19], and NBeats [12], open up a realm of data-driven predictions for extreme events without necessitating a comprehensive understanding of the system's dynamics. These data-centric models are widely employed to explore the emergence of extreme climatic events through training over a significantly large dataset. While all these methods share the common goal of delivering precise short-term forecasts by capturing long-term system trajectories, they do not explicitly account for the physical laws that inherently govern these chaotic systems.

Recently, the integration of physical knowledge into data-driven models has received significant attention among researchers for their ability to model real-world phenomena by addressing the everlasting challenges of modeling chaotic systems. Physics-informed neural networks (PINN) [14, 8] integrates prior physics knowledge within data-driven neural networks using a series of differential equations. The physical knowledge of the PINN models enables them to enhance training data, modify the architectural design of the network, and perform physics-informed optimization depending on the location of the knowledge integration. In recent studies, PINN-based approaches have been widely used as a function approximator in various applications such as modeling 3D temperature data using physical laws [24], simultaneous prediction of observed and unobserved variables in chaotic systems [13], predicting the lake surface temperature [2], and predicting the effect of cloud processes on climate [1].

This work aims to enhance the forecasting ability of the LSTM networks for extreme climatic events dataset by embedding the physical laws of the Van der Pol Oscillator into the framework. The proposed Van der Pol-informed Neural Networks (VPINN) approach learns the dynamics of the nonlinear oscillator and integrates the learned representations with the temporal dynamics of the data to generate a multi-step ahead forecast of the desired horizon.

## 1.1   Contributions of this work

VPINN is introduced to model the temporal evolution of real-world extreme climatic events, including seismic waves, sea surface temperature, humidity, wind speed, and temperature for varied locations. We obtain the ground truth data from various open sources and also simulate a dataset from the Van der Pol oscillator, as outlined in Section 2. We then infuse the dynamics of this oscillator into a conventional LSTM framework, both through a transfer learning mechanism and the utilization of a physics-informed loss function for capturing the complex patterns inherent in the extreme events time series data. In contrast to the statistical models such as exponential smoothing (SES) [6] and the deep learning models, the VPINN approach exhibits the capability to learn the complex behavior of the data and is also suitable for handling limited data problems.

Our experimental results demonstrate that the proposed VPINN outperforms existing state-of-the-art models for multi-step forecasts across various time horizons in majority scenarios. Notably, we observe a significant improvement in forecasting accuracy, with the VPINN model enhancing the performance of the LSTM network by 53.47% due to the integration of the physical laws in the model. These simple yet essential experiments underscore the importance of integrating physics-informed forecasting techniques, such as the VPINN framework, into the existing data-driven climate modeling paradigms.

## 2   Preliminaries

**Van der Pol system:** Nonlinear oscillator systems have applications across a broad spectrum of physical phenomena, spanning atmospheric physics, nonlinear optics, plasma physics, electronics, biophysics, and chemical reactions, among many others [18]. These systems are characterized by their nonlinearity, which often leads to complex and chaotic behavior. They exhibit multiple equilibria, modulation in amplitude and frequency, and sensitivity to initial conditions, making them crucial in modeling the complexity and diversity of real-world phenomena. For instance, Van der Pol oscillator systems serve as a valuable tool for the study of extreme events. The Van der Pol equation, used in modeling nonlinear dynamical systems, is a non-conservative self-oscillatory system with nonlinear damping [21]. This mathematical formulation evolves in time following a second-order differential

equation, taking the following form:

$$\frac{d^2x}{dt^2} - \mu(1 - x^2)\frac{dx}{dt} + x = 0, \tag{1}$$

where the position coordinate $x$ is a function of time $t$ and the damping strength of the oscillation is expressed using a positive scalar parameter $\mu$. The Van der Pol system has a unique stable limit cycle, i.e., when time is close to infinity all nearby solutions of Eq. 1 tend towards a periodic solution. Moreover, the Van der Pol oscillator exhibits a chaotic dynamical nature following [11], which allows this dynamical system to efficiently model chaotic datasets.

**Proposition 1.** *The non-conservative Van der Pol oscillator's dynamics evolving in time is chaotic.*

*Proof.* The state-space equation of the Van der Pol oscillator (Eq. 1) can be expressed as,

$$\frac{d\psi_1}{dt} = \psi_2 \quad \text{and} \quad \frac{d\psi_2}{dt} = -\psi_1 - \mu\psi_2\left(1 - {\psi_1}^2\right), \tag{2}$$

where $\psi_1 = x$ and $\psi_2 = \frac{dx}{dt}$. In vector notation Eq. 2 can be represented as:

$$\frac{d\Psi(t)}{dt} = \mathcal{H}(\psi(t), \mu), \tag{3}$$

where, $\Psi = [\ \psi_1(t) \quad \psi_2(t)\ ]^T$ is the space vector and $\mathcal{H} = [\ \mathcal{H}_1 \quad \mathcal{H}_2\ ]^T$ is the coefficient vector. The dynamics of Eq. 3 when subjected to small deviations from the defined trajectory would be:

$$\delta\left(\frac{d\Psi(t)}{dt}\right) = \mathcal{L}_{i,j}(\Psi(t))\,\delta\Psi; \quad i, j = 1, 2,$$

where $\mathcal{L}_{i,j} = \frac{\partial\mathcal{H}_i}{\partial\psi_j}$ is the Jacobian Matrix, comprising of derivatives. The chaotic behavior of the dynamical system can be inferred based on the positive value of the maximal Lyapunov exponent. Following, [23] the maximal Lyapunov exponent of the system can be defined,

$$\lambda_{\max} = \lim_{t \to \infty} \frac{1}{t} \log \frac{\|\delta\Psi(t)\|}{\|\delta\Psi(0)\|}.$$

By utilizing the Runge-Kutta method of order 4, [11] showed that $\lambda_{\max} \approx 0.095\ (>0)$. Hence, the dynamics of the Van der Pol oscillator are chaotic in nature.

$\square$

**Long-Short Term Memory (LSTM):** LSTM networks represent a modification of classical Recurrent Neural Networks (RNNs) designed to address the vanishing gradient problem, thereby enhancing training stability [7]. These networks are widely employed in various sequential learning tasks, including natural language processing, machine translation, image captioning, and time series analysis. Their distinctive chain-like architecture consists of three key components: the *input gate*, *output gate*, and *forget gate*. This gating mechanism regulates the information flow within the cell state, serving as the long-term memory storage and the hidden state, representing its short-term counterpart. Given the input vector (say $x_i$) at the $i^{th}$ time step, the forget gate determines how much information from the previous hidden state $\tilde{h}_{i-1}$ should be retained at time $i$. It uses a sigmoidal activation function over a weighted combination of $x_i$ and $\tilde{h}_{i-1}$. Thus the resulting activation vector $F_i$, indicating how much information to forget or keep obtained as:

$$F_i = \sigma\left(W_1^x x_i + W_1^{\tilde{h}} \tilde{h}_{i-1} + b_1\right),$$

with $W_1^x, W_1^{\tilde{h}}$ as the weights and $b_1$ as the bias. The input gate, on the other hand, utilizes the sigmoidal and tanh activation functions to update the cell state with the current input. The two activation vectors of the input gate are computed as:

$$I_i = \sigma\left(W_2^x x_i + W_2^{\tilde{h}} \tilde{h}_{i-1} + b_2\right) \quad \text{and} \quad g_i = \tanh\left(W_3^x x_i + W_3^{\tilde{h}} \tilde{h}_{i-1} + b_3\right),$$

where $W$ and $b$ indicate the weights and bias respectively. The current cell state $C_i$ is then calculated by combining the output from the forget gate and the input gate as:

$$C_i = F_i \odot C_{i-1} \oplus I_i \odot g_i,$$

where $\odot$ is the point-wise multiplication and $\oplus$ is the direct sum operator. This combination ensures both long-term and short-term memory components are appropriately considered. In the output gate, the current hidden state is updated using sigmoidal and tanh activation functions. These activations determine the new hidden state as:

$$O_i = \sigma \left( W_4^x x_i + W_4^{\tilde{h}} \tilde{h}_{i-1} + b_4 \right) \quad \text{and} \quad \tilde{h}_i = O_i \odot \tanh \left( C_i \right).$$

Finally, $\tilde{h}_i$ is used to compute the output at the current time step as $\hat{y}_i = \sigma \left( \tilde{W} \tilde{h}_i + b_5 \right)$. Overall, LSTM's robust long-term memory retention capabilities make them valuable for modeling the complexities of real-world phenomena [22].

## 3  Proposed Approach

The proposed VPINN framework seamlessly integrates the information from both the real measurements and the chaotic dynamics of the Van der Pol oscillator. This model leverages the transfer learning approach and physics-based regularized loss function to significantly improve the modeling and forecasting capabilities of the LSTM network. Training the sequential structure of the LSTMs can be computationally intensive and time-consuming. To address this challenge, we provide prior knowledge via pre-training to our VPINN model in a task-agnostic manner. Our proposed architecture, as illustrated in Figure 1, is a sequential approach that learns through a combination of transfer learning and data-driven learning. In the first phase, we generate a synthetic dataset by simulating data points from the nonlinear Van der Pol oscillator (as in Eq. 1) with $\mu = 4$ using the Runge-Kutta method. The second phase of the architecture involves training a standard LSTM network [7] on both the real-time series and the time derivatives of the simulated series while enforcing the physical law as a regularization term in the network. The proposed framework aims to learn the complex patterns and the chaotic behavior of the data-generating mechanism using historical values and time derivatives. To compute the physics-based regularization term, we follow the transductive PINN model [14], where time-indexed inputs are provided to a regularized multi-layered perceptron to generate the solution of the differential equation as the output. The regularization term in PINN amounts to differentiating the network and computing the time derivatives using automatic differentiation. Since real-world extreme event datasets consist of discrete observations, it becomes challenging to use automatic differentiation for computing the time derivatives. To mitigate this issue, we compute the discrete derivatives of the time series using the First Principle of Derivatives. Thus for a simulated time series $x(t)$ indexed at time $t$, we compute the discrete-time derivatives as:

$$\frac{dx}{dt} = \frac{x\left(t + \delta t\right) - x(t)}{\delta t}, \tag{4}$$

where $\delta t$ is the time lag. In our framework, we set $\delta t = 1$ since real-world time series datasets are recorded chronologically in time. Once the simulated data is generated, the VPINN network receives the tuple $y_{\text{Real}}^{(t)} = \left\{ y_t^*, \frac{dx}{dt}, \frac{d^2x}{dt^2} \right\}$ as input, where $y_t^*$ represents the training data value and $\frac{dx}{dt}, \frac{d^2x}{dt^2}$ are the first-order and second-order time derivatives of $x(t)$ computed using Eq. 4, respectively. The network aims to predict the values of the subsequent time steps in a multivariate setting. The physical dynamics of the Van der Pol oscillator are imposed on the proposed model through both transfer learning and the introduction of a physics-based loss function. To compute a quantifiable measure of physical consistency within the model, the predicted values $\hat{y}_{\text{Pred}}^{(t)} = \left\{ \hat{y}_t^*, \frac{\hat{dx}}{dt}, \frac{\hat{d^2x}}{dt^2} \right\}$ have to satisfy the Van der Pol equation. Therefore, based on Eq. 1, the physics-based loss function for enforcing the dynamics of the Van der Pol oscillator on the predicted values can be calculated as follows:

$$\text{Loss}^{\text{Phy}} = \frac{d^2 \hat{y}_{\text{Pred}}^{(t)}}{dt^2} - \mu \left( \frac{d\hat{y}_{\text{Pred}}^{(t)}}{dt} - \left( \hat{y}_{\text{Pred}}^{(t)} \right)^2 \frac{d\hat{y}_{\text{Pred}}^{(t)}}{dt} - \frac{\hat{y}_{\text{Pred}}^{(t)}}{\mu} \right). \tag{5}$$

This physical information is integrated into the objective function of our proposed VPINN model through a modification of the conventional loss function, $\text{Loss}^{\text{Data}}$, which is calculated using the model predictions and the true output labels as

$$\text{Loss}^{\text{Data}} = \text{RMSE} \left( y_t^*, \hat{y}_t^* \right), \tag{6}$$

where RMSE is the Root Mean Square Error. In contrast, the modified loss function includes the additional physics-based loss, formulated using Eq. 6 and Eq. 5 as follows:

$$\text{Loss}^{\text{Total}} = \text{Loss}^{\text{Data}} + \lambda_{\text{Phy}} \text{Loss}^{\text{Phy}}, \tag{7}$$

where $\lambda_{\text{Phy}}$ represents the hyperparameter corresponding to the physics-based loss function. The Eq. 7 is designed to enhance the model's generalization performance by simultaneously optimizing accuracy and ensuring physical consistency. Since the final loss function is nearly differentiable everywhere, we employ the backpropagation algorithm to calculate and propagate gradients across various layers. For visualizing the working principle of our proposed VPINN model, a detailed architecture of the model is presented in Figure 1.

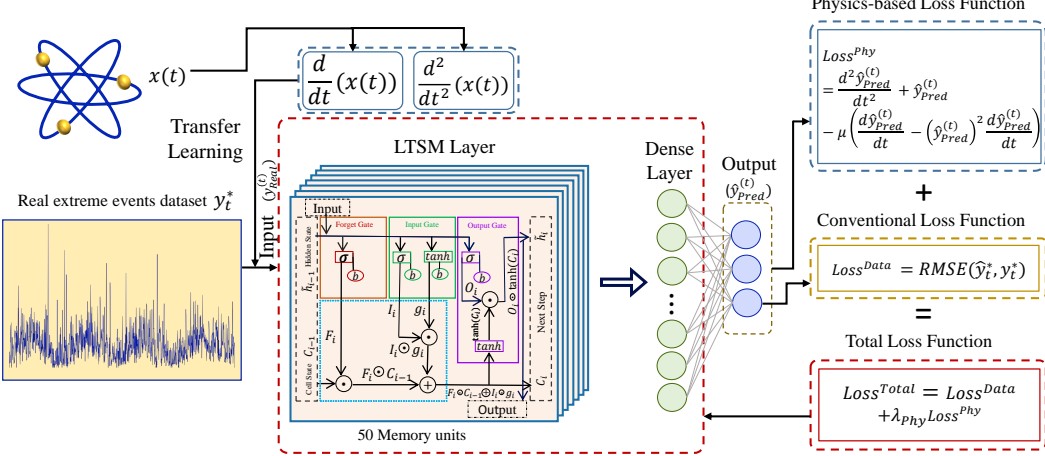

Figure 1: **Van der Pol-informed neural networks (VPINN).** We generate simulated data from the Van der Pol oscillator and calculate its time derivatives. We concatenate these derivatives with the real target series and model them using LSTM layers and a dense layer to generate the subsequent predictions. A modified loss function, combining the conventional loss and the physics-based loss, is used to train the network with a backpropagation approach.

## 4 Experimental Setup and Results

**Dataset.** To ensure a fair comparison and evaluation of the proposed VPINN framework, we employ a set of real-world extreme climatic event datasets with varying dynamics. We selected five time series datasets with diverse temporal units to demonstrate the scalability of our approach. These publicly available datasets include [1]: (1) *Turkey Seismic Waves*, (2) *El Niño Sea Surface Temperature (SST)*, (3) *Philippines Temperature*, (4) *Madrid Humidity*, and (5) *Delhi Wind Speed*. Each of these datasets features varied climatic events and their anomalous behavior has the potential to significantly disrupt the global climatic patterns. A brief overview of these datasets along with their statistical properties and forecast horizons is presented in Table 1. All the experiments conducted in this study encompass both short-term and long-term time series forecasting settings for these datasets.

**Performance Indicators.** In our study we assess the effectiveness of our proposed approach by employing two commonly used evaluation metrics: *Root Mean Square Error* defined as $\text{RMSE}(y^*, \hat{y}) = \sqrt{\sum_{t=1}^{n} (y_t^* - \hat{y}_t)^2 / n}$, and *Mean Absolute Error* computed as $\text{MAE}(y^*, \hat{y}) = \sqrt{\sum_{t=1}^{n} |y_t^* - \hat{y}_t| / n}$, where $y^*$ denote the ground truth observations and $\hat{y}$ indicate the corresponding predicted value for $n$ time-steps. These evaluation metrics are widely recognized and are commonly used in the context of extreme events forecasting problems [15].

**Experimental Results.** Table 2 presents the main experimental results of the VPINN framework for both short-term and long-term time series forecasting tasks. Since our proposal and the baseline

---

[1] www.wunderground.com/; www.kaggle.com/; www.ncei.noaa.gov/

Table 1: Extreme events data characteristics. The green circles represent the presence of the feature, while the red one resembles its absence. The forecast horizons used for these datasets depend on their respective temporal unit.

| Dataset | Granularity | Start date | Time steps | Statistical properties | | | | Forecast horizon | |
|---|---|---|---|---|---|---|---|---|---|
| | | | | Stationary | Linear | Chaotic | Seasonal | Short-term | Long-term |
| Turkey Seismic Waves | 1 day | 18/01/2000 | 6574 | 🔴 | 🔴 | 🔴 | 🔴 | 10 days | 30 days |
| El Niño SST | 1 week | 03/01/1990 | 1634 | 🟢 | 🔴 | 🟢 | 🔴 | 13 weeks | 26 weeks |
| Philippines Temperature | 1 hour | 07/02/2012 | 1488 | 🔴 | 🟢 | 🔴 | 🟢 | 24 hours | 48 hours |
| Madrid Humidity | 1 hour | 01/01/2019 | 2447 | 🔴 | 🔴 | 🟢 | 🟢 | 24 hours | 48 hours |
| Delhi Wind Speed | 1 day | 01/01/2013 | 1462 | 🟢 | 🔴 | 🟢 | 🟢 | 10 days | 30 days |

| Model | | SES [6] | | LSTM [7] | | RCN [16] | | Prophet [19] | | Bi-LSTM [17] | | NBeats [12] | | VPINN | | IMP |
|---|---|---|---|---|---|---|---|---|---|---|---|---|---|---|---|---|
| Metric | | RMSE | MAE | RMSE | MAE | RMSE | MAE | RMSE | MAE | RMSE | MAE | RMSE | MAE | RMSE | MAE | RMSE |
| Turkey | ST | 0.555 | 0.506 | 2.912 | 2.014 | 0.576 | 0.381 | 0.576 | 0.432 | 3.630 | 3.522 | 0.573 | 0.431 | **0.514** | **0.360** | 82.3% |
| | | (0.00) | (0.00) | (0.08) | (0.02) | (0.12) | (0.10) | (0.10) | (0.01) | (0.10) | (0.12) | (0.01) | (0.10) | (0.01) | (0.01) | |
| SW | LT | 1.150 | 1.108 | 2.752 | 1.879 | 1.332 | 0.617 | 0.553 | 0.389 | 3.544 | 3.429 | 0.552 | 0.504 | **0.446** | **0.323** | 83.7% |
| | | (0.00) | (0.00) | (0.01) | (0.01) | (0.99) | (0.52) | (0.08) | (0.05) | (0.08) | (0.15) | (0.01) | (0.01) | (0.01) | (0.01) | |
| Delhi | ST | 14.75 | 14.52 | 8.026 | 6.285 | 32.95 | 29.83 | 6.079 | 4.872 | 7.982 | 6.693 | 5.712 | **4.648** | **5.604** | 4.863 | 30.2% |
| | | (0.00) | (0.00) | (0.83) | (0.89) | (5.31) | (6.17) | (2.72) | (1.10) | (0.81) | (1.23) | (1.00) | (0.67) | (0.28) | (0.39) | |
| WS | LT | 8.040 | 7.532 | 6.748 | 4.812 | 36.88 | 26.38 | 6.726 | 5.258 | 6.769 | 5.167 | 6.749 | 4.914 | **6.673** | **4.369** | 1.11% |
| | | (0.00) | (0.00) | (0.92) | (0.85) | (8.92) | (5.38) | (1.21) | (0.97) | (1.00) | (0.95) | (1.01) | (0.45) | (0.21) | (0.20) | |
| El Niño | ST | 18.08 | 18.06 | 22.38 | 22.22 | 5.352 | 4.182 | **2.346** | **1.924** | 23.32 | 23.26 | 2.934 | 2.669 | 7.471 | 7.208 | 66.6% |
| | | (0.00) | (0.00) | (1.05) | (1.61) | (2.01) | (1.28) | (1.00) | (0.99) | (1.27) | (0.95) | (0.18) | (0.98) | (0.43) | (0.48) | |
| SST | LT | 22.65 | 22.55 | 19.82 | 19.69 | 15.15 | 12.19 | 7.201 | 6.841 | 21.57 | 21.44 | 7.273 | 6.385 | **6.016** | **5.426** | 69.6% |
| | | (0.00) | (0.00) | (4.64) | (1.45) | (3.99) | (2.31) | (1.27) | (1.56) | (1.98) | (1.46) | (1.72) | (0.55) | (0.34) | (0.35) | |
| Madrid | ST | 63.09 | 61.01 | 46.47 | 44.28 | 26.98 | 24.49 | 28.52 | 25.79 | 48.79 | 46.72 | 27.73 | 22.61 | **26.14** | **22.04** | 43.7% |
| | | (0.00) | (0.00) | (2.52) | (2.47) | (3.01) | (9.48) | (2.16) | (1.99) | (1.98) | (1.37) | (1.99) | (2.89) | (0.01) | (0.00) | |
| Humidity | LT | 67.60 | 65.84 | 52.78 | 50.83 | 61.90 | 52.92 | 34.89 | **26.89** | 54.23 | 52.35 | **31.64** | 27.69 | 35.76 | 32.75 | 32.2% |
| | | (0.00) | (0.00) | (2.89) | (1.89) | (1.76) | (1.95) | (2.20) | (2.09) | (2.10) | (2.51) | (1.54) | (2.74) | (0.01) | (0.01) | |
| Philippines | ST | 17.14 | 16.83 | 25.01 | 24.68 | 18.78 | 16.78 | 14.57 | 13.98 | 26.06 | 25.76 | 13.95 | 13.56 | **13.89** | **13.29** | 44.4% |
| | | (0.00) | (0.00) | (1.74) | (1.85) | (2.97) | (1.89) | (1.78) | (1.09) | (1.30) | (1.99) | (2.01) | (2.00) | (0.99) | (2.30) | |
| Temp | LT | 16.57 | 16.12 | 26.04 | 25.70 | 17.05 | 15.16 | 18.25 | 17.86 | 26.46 | 26.13 | 20.44 | 19.69 | **12.94** | **12.56** | 50.3% |
| | | (0.00) | (0.00) | (1.92) | (2.08) | (2.08) | (2.01) | (1.99) | (0.18) | (2.10) | (2.23) | (2.27) | (1.92) | (1.25) | (2.01) | |

Table 2: Short-term and long-term forecasting performance (RMSE and MAE) of the proposed VPINN model in comparison to the state-of-the-art forecasting techniques (best results are **highlighted**).

forecasters used in this experiment depend on previous observations, we set the same lag length $k$ as 3.5 times the desired forecast horizon $h$. Thus $k$-prior observations of the target series are fed into the forecasting algorithm for predicting $h$ subsequent values of the series. Additionally, to efficiently train the models we divide the available training data (after removing the test observations) into train (80%) and validation (20%) sets. In our proposed architecture, the LSTM layer with 50 hidden units is followed by a dense layer with $h$ units. To ensure a fair comparison we use the same architecture for the LSTM model as that of the proposal. In order to quantify the variance of the data-driven forecasters we repeat the experiments 10 times with random initializations and report the mean RMSE and mean MAE scores with their corresponding standard deviations for both short-term and long-term forecasting tasks. As can be observed in Table 2, the proposed VPINN framework achieves state-of-the-art performances in most benchmarks and forecast horizon settings. Overall, the proposal yields a 53.47% improvement in terms of RMSE scores, compared to the conventional LSTM network. Moreover, we observe that the traditional SES model exhibits poor performance for this task owing to its inability to handle the nonlinearity of the datasets. On the other hand, deep learning models like NBeats, and Prophet being fed with a substantial amount of training data are able to capture the complex dynamics of the real-world extreme events datasets. However, their overall forecasting performance significantly lags behind the proposed VPINN framework. In the case of long-term forecasting tasks, our proposal improves the LSTM model by 47.42%. We attribute this accuracy enhancement to the proposed VPINN's ability to model the chaotic nature of the datasets along with other complexities due to the prior knowledge from pre-training and physical regularized loss function. To demonstrate an effective visual comparison of the state-of-the-art models and the proposed framework, we present the step-wise RMSE metric computed sequentially by increasing the forecast horizon for selected datasets in Fig. 2.

For statistical comparison of the benchmarks, we conducted the Multiple Comparisons with the Best (MCB) test [3] to assess the disparities in forecast skills. Fig. 3 summarizes the results of the MCB test along with the forecasters' ranks based on the RMSE and MAE metrics. The figure reveals that the proposed VPINN framework achieves a minimum rank of 1.50 and 1.60 w.r.t. to RMSE and

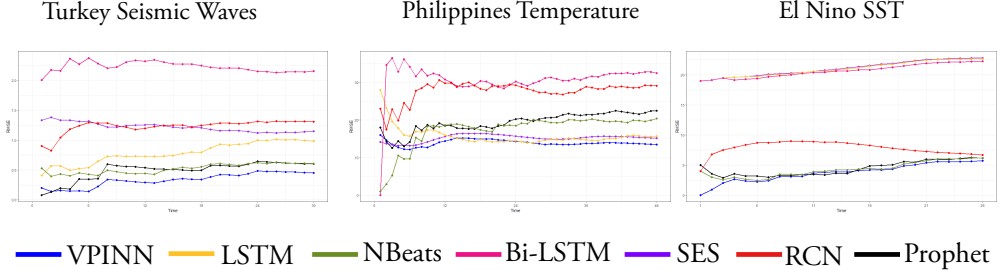

Figure 2: Forecast accuracy of the proposed model and the state-of-the-art for selected datasets. The images provide a comparison of the RMSE metric computed at each forecast step.

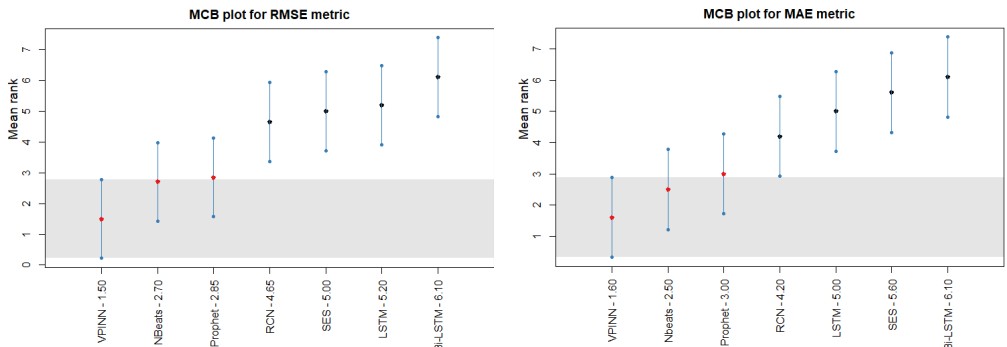

Figure 3: Visualization of the MCB analysis w.r.t. RMSE (left) and MAE (right) metric. The Y-axis of the plot shows the average rank and the X-axis represents the corresponding model.

MAE scores, respectively, indicating its overall "best" performance compared to other models such as NBeats and Prophet. This enhanced forecasting capability of the VPINN model is attributed to its hybrid approach, which allows it to leverage both time series data and associated physics knowledge acquired through prior training via transfer learning.

## 5 Discussion and Conclusion

We proposed a physics-informed forecasting model, namely VPINN, by inducing the physical dynamics of the Van der Pol oscillator into the data-driven LSTM network. Compared to state-of-the-art deep learners, VPINN enhances generalization through a combination of transfer learning and a physics-informed loss function. The modeling capabilities of the VPINN framework show promise for developing and refining additional physics-guided forecasters capable of handling the complex geophysical turbulence of extreme climatic events. Furthermore, exploring the integration of other nonlinear dynamical systems into machine learning and deep learning frameworks for tackling more complex geophysical challenges requires further investigation. This work will act as the middle ground between domain-specific knowledge and pure data-driven methods. However, the choice of physical laws for real-world applied problems plays a critical role in the proposal and may degrade the performance of our architecture. We plan to take these issues into consideration in our future research

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
