# OpenReview forum: "Van der Pol-informed Neural Networks for Multi-step-ahead Forecasting of Extreme Climatic Events"
_NeurIPS.cc/2023/Workshop/AI4Science — NeurIPS2023-AI4Science Poster_

### Official Review · Reviewer_1wMN · 2023-10-08
**Overall good paper**

**Rating:** 6
**Confidence:** 4

**Review:**

The authors proposed a new physics-informed neural network for extreme event prediction. The benchmark shows the new model's superior performance. Here are several review suggestions:

1. The tested dataset (i.e., SST, humidity) is not for extreme events. The performance metrics (MSE, MAE) does not emphasize the extreme values. So I don't think the title is proper here.
2. Figure 2 is hard to read. The font size should be larger.
3. Is the synthetic data from the Van Der Pol generator independent of the task dataset? Is there any other oscillator system that can be used?

The overall paper is good.

---

### Meta-Review · Area_Chair_GnJF · 2023-10-27

**Recommendation:** Accept (Poster)
**Confidence:** 4

**Metareview:**

This paper proposes a new type of PINN that demonstrates superior performance in weather event forecasting, outperforming traditional time series forecasting models. Overall, this paper delivers valuable insights for both the PINN and weather time series communities. However, relying solely on the Van der Pol model might not fully encompass all the non-linearities inherent in climatic events, though it serves as a commendable starting point. Additionally, in light of the review, it is advisable to moderate the title. In the introduction, several recent forecasting paper are missing, such as: https://arxiv.org/abs/2207.05833 (earthformer), https://arxiv.org/abs/2307.10422 (prediff)